# Export of piRNA precursors by EJC triggers assembly of cytoplasmic Yb-body in *Drosophila*

Cynthia Dennis[1], Emilie Brasset[1], Arpita Sarkar[1] & Chantal Vaury[1]

PIWI-interacting RNAs (piRNAs) are effectors of transposable element (TE) silencing in the reproductive apparatus. In *Drosophila* ovarian somatic cells, piRNAs arise from longer single-stranded RNA precursors that are processed in the cytoplasm presumably within the Yb-bodies. piRNA precursors encoded by the *flamenco* (*flam*) piRNA cluster accumulate in a single focus away from their sites of transcription. In this study, we identify the exportin complex containing Nxf1 and Nxt1 as required for *flam* precursor nuclear export. Together with components of the exon junction complex (EJC), it is necessary for the efficient transfer of *flam* precursors away from their site of transcription. Indeed, depletion of these components greatly affects *flam* intra-nuclear transit. Moreover, we show that Yb-body assembly is dependent on the nucleo-cytoplasmic export of *flam* transcripts. These results suggest that somatic piRNA precursors are thus required for the assembly of the cytoplasmic transposon silencing machinery.

[1] Univ Clermont Auvergne, CNRS, INSERM, GReD, BP 10448, F-63001 Clermont-Ferrand, France. Correspondence and requests for materials should be addressed to C.V. (email: chantal.vaury@udamail.fr).

Piwi-interacting RNAs (piRNAs) are a class of 23–29 nucleotide (nt) small RNAs that are expressed in animal gonads. In the metazoan germline, they play an important role in silencing transposable elements (TEs) and are thus critical for genome integrity[1–3]. In *Drosophila*, piRNAs originate from the processing of piRNA precursors transcribed from discrete genomic regions called piRNA clusters. piRNA clusters are enriched in TEs or their vestiges. In *Drosophila melanogaster*, their size ranges from 1 to a few hundred kb and they are almost exclusively located within or in close proximity to heterochromatic regions[1]. The ovarian somatic follicle cells preferentially express uni-strand piRNA clusters, transcribing one strand only. Current knowledge on piRNA cluster biology has been mostly acquired from studies conducted on *flam* locus, a uni-strand piRNA cluster active in the follicular cells[4]. *flam* is transcribed by the RNA polymerase II (RNA pol II) into a precursor that is similar to other RNA Pol II transcripts in basic RNA features[5,6]. Before being processed into piRNAs, the presumably long single-stranded transcript is spliced. Indeed, the *flam* precursor has been shown to contain a constitutive first exon. Several alternative splicings have also been reported[5].

In the follicle cells, RNA precursors transcribed from piRNA clusters are first transferred to cytoplasmic perinuclear granules called Yb-bodies that have been reported to be major sites for piRNA biogenesis[7,8]. RNA precursors are then processed into primary piRNAs whose 5′ ends are defined by the endonucleolytic cleavage of Zucchini (Zuc) endonuclease[9,10]. They are loaded onto the PIWI clade protein, Piwi, and translocate to the nucleus where they elicit transcriptional silencing[11–16].

In a previous study, Dennis *et al.*[17] found that *flam* precursors channel through the nucleoplasm and first accumulate in a structure called Dot COM within the nucleus. This nuclear structure faces the cytoplasmic Yb-body and has been proposed to be a nuclear site of accumulation of piRNA precursors coming from various piRNA clusters before their export. This nuclear accumulation was not detected by Murota *et al.*, who instead reported a cytoplasmic *flam* focus[18]. Whatever the positioning of this first accumulation of *flam* transcripts before processing, it is still unknown how such a transcript can be distinguished from other mRNAs and find its way to the Yb-bodies. A sequence crucial for piRNA production has been identified[19]. When a construct containing the 5′ end of *flam* fused to *luciferase* reporter gene is transfected into the ovarian somatic cells, *luciferase* RNA is processed via the primary processing machinery, giving rise to thousands of piRNAs. This prompted the authors to suggest that the 5′ end of *flam* transcript is a potential piRNA-trigger sequence. A *cis*-acting 100-nt-long fragment bound by Yb—a core component of Yb-bodies, has also been identified in the *traffic jam* (Tj) 3′ untranslated region that is sufficient for producing artificial piRNAs from unintegrated DNA[20].

Several screens performed on *Drosophila* revealed factors required for the piRNA pathway[21–23]. The Nxf1/Tap and Nxt1/p15 export proteins as well as components of the exon junction complex (EJC) and UAP56 protein have emerged from these screens. Nxf1, a member of the NXF family of transport receptors, and its cofactor Nxt1, a low-molecular-weight NTF2-like protein, are thought to mediate export of the majority of cellular mRNAs[24–27]. Nxt1 enhances Nxf1-dependent RNA export by stimulating Nxf1 interactions with the nuclear pore complex[28,29]. Binding of the Nxf1–Nxt1 heterodimer to mRNAs is thought to be mediated by adaptor proteins. Consistently, EJC has been shown to facilitate the recruitment of Nxf1 to spliced mRNAs[30]. EJC is deposited onto mRNA upstream of exon–exon junctions concomitantly to splicing and remains associated during nuclear export[25]. It works as a platform for various transiently interacting factors, such as Nxf1–Nxt1 complex and the nuclear mRNA export factor UAP56, to link together several mRNA processes such as transport, translation or stability[31–34]. The two EJC core factors Mago and Tsunagi/Y14 (Tsu) as well as the EJC accessory proteins RnpS1 and Acinus (Acn) have been implicated in the piRNA pathway biology, their mutations leading to TE derepression[21,23,35–37].

In the present study, we identify exportins-Nxt1 and Nxf1, EJC components—Mago, Tsu, RnpS1, Acn and UAP56 proteins—as factors required at different steps of *flam* nuclear traffic and export to the cytoplasm. We also demonstrate that export of piRNA precursors is the signal for the Yb-bodies to be assembled. In the follicle cells the *flam* cluster, which gives rise to most piRNAs, acts as a master locus for the formation of Yb-bodies.

## Results

**Localization of piRNA precursors before processing.** We first investigated the nucleo-cytoplasmic distribution of *flam* transcripts in more than 1,700 follicular cells. *flam* transcripts were revealed by *in situ* hybridization using the specific *flam* 508 RNA probe that recognizes a unique sequence of the *flam* locus[17]. An anti-lamin antibody was used to label the nuclear periphery and decipher the subcellular localization of *flam* transcripts in wild-type (WT) flies *ISO1A*. As previously reported, *flam* precursor transcripts were detected as a single focus per cell although a few cells displayed one or two foci at early stages of oogenesis[17,18]. This focus was always in close proximity to or at the nuclear periphery. 46% of cells (n = 1,764) had a focus co-localizing with the lamin staining (Fig. 1, three middle panels and Fig. 2a, middle panel). In a few cases, *flam* staining seemed to extend through the lamin signal (Fig. 1, third and fourth panels). In all, 15% and 39% of cells had *flam* transcripts within the nucleus (a focus previously named Dot COM) and the cytoplasm, respectively (Fig. 1, left and right, respectively; Fig. 2a, left and right; Supplementary Movie 1). These findings suggest that most *flam* transcripts gather together

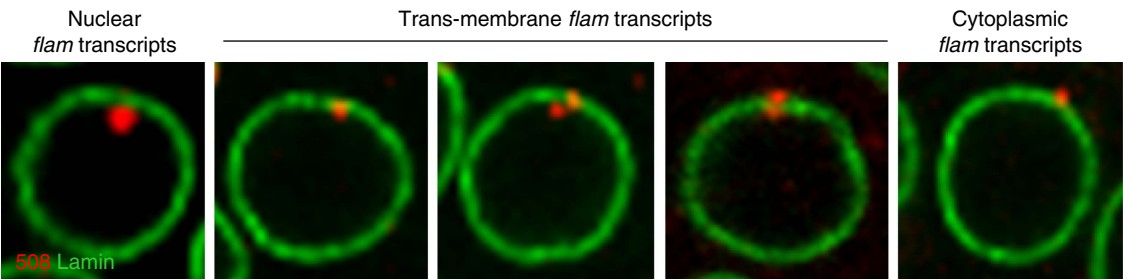

| Nuclear *flam* transcripts | Trans-membrane *flam* transcripts | | | Cytoplasmic *flam* transcripts |

**Figure 1 | Sub-cellular localization of *flam* piRNA precursor transcripts in ovarian follicle cells.** Examples of subcellular localization of *flam* precursors in ovarian follicle cells. *flam* transcripts revealed by *flam* 508 RNA probe are in red and nuclear periphery revealed with anti-lamin antibody is visualized in green.

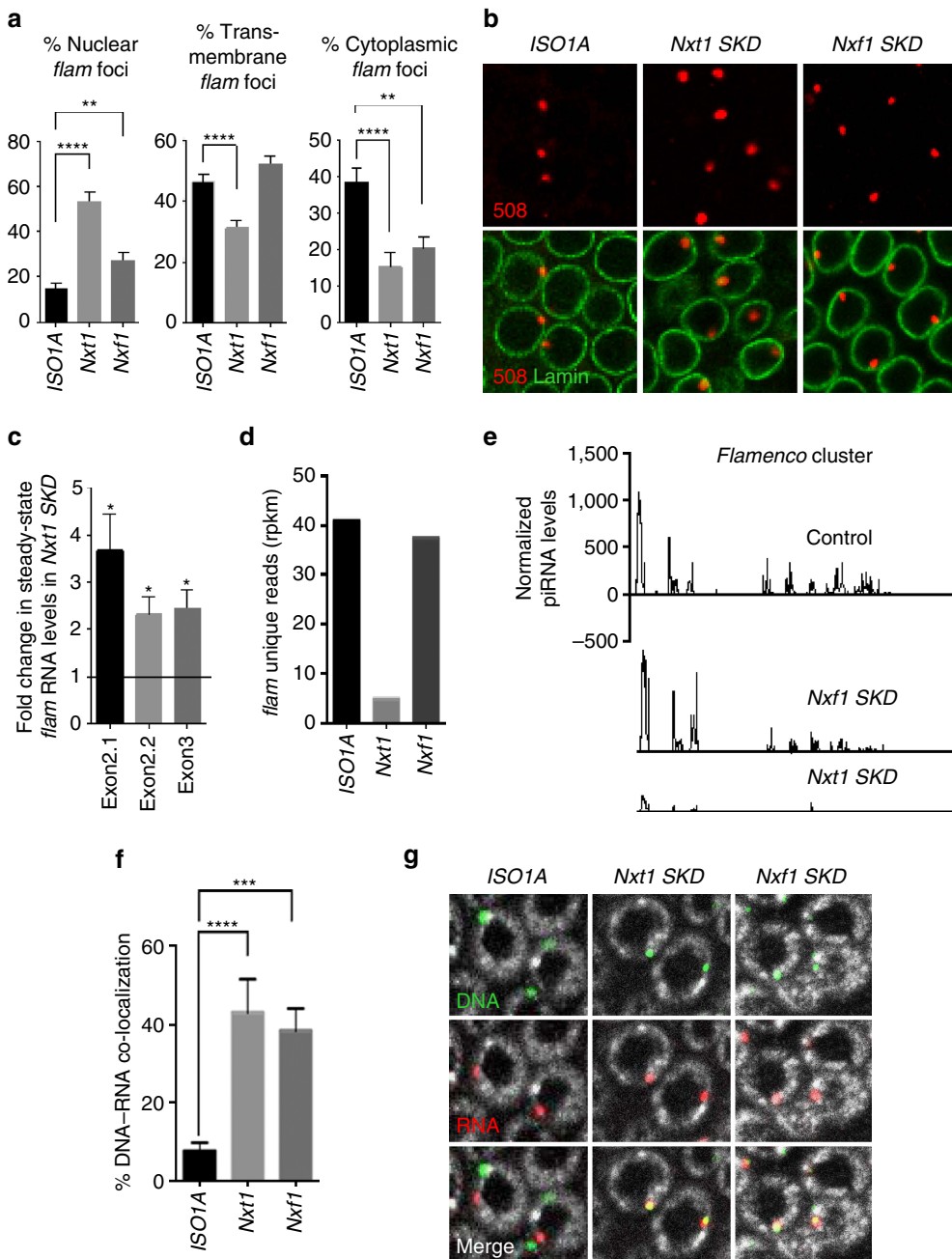

**Figure 2 | Exportin Nxt1 and Nxf1 are involved in *flam* RNA export and intra-nuclear traffic to Dot COM.** (**a**) Quantitative analysis of *flam* transcript localization in ovarian follicle cells. The percentage of cells with a nuclear (left), transmembrane (middle) and cytoplasmic (right) focus is plotted. Error bars represent s.e.m. $n = 45$ (*ISO1A*), $n = 28$ (*Nxt1*), $n = 22$ (*Nxf1*), $n$ is the number of independent experiments. ****$P$ value $< 0.0001$ and **$P$ value $< 0.01$ are calculated according to Mann–Whitney test. (**b**) *flam* transcripts (red) and nuclear membrane (green) are visualized using *flam* 508 RNA probe and anti-lamin antibody, respectively. One $Z$-stack of confocal image is shown. Therefore, as *flam* foci are not all in the same focus they cannot be all visualized in every nucleus of the field. (**c**) Fold change in steady-state RNA level of *flam* in *Nxt1-SKD* ovaries compared with *ISO1A* ovaries. RNA levels are normalized to *rp49* level. The positions of the primers used are indicated above and their sequence is given in Supplementary Table 2. Data are presented as means ($n = 3$). Error bars indicate s.e.m. *$P$ value $< 0.05$, according to the Student's $t$-test. (**d,e**) Changes in steady-state levels of *flam* unique 23–29 nt piRNAs in ovarian cells of *ISO1A* and *Nxt1-* and *Nxf1-SKD* lines measured by small RNA sequencing (normalized to one million genome-mappable reads). (**d**) Density plot of unique 23–29 nt piRNAs mapping to the *flam* piRNA cluster. The $Y$ and $X$ axes in **e** are identical for the three graphs. $X$ axis represents the *flam* 180 kb starting from its transcription start site on the left. $Y$ axis represents the quantity of piRNAs normalized to one million genome mappable reads. (**f**) Quantitative analysis of relative localization between *flam* DNA and *flam* transcripts in follicle cells. The percentage of cells with a DNA–RNA co-localization is plotted. Error bars represent s.e.m. $n = 32$ (*ISO1A*), $n = 20$ (*Nxt1*), $n = 38$ (*Nxf1*), $n$ is the number of independent experiments. ****$P$ value $< 0.0001$, ***$P$ value $< 0.001$), according to Mann–Whitney test. Error bars represent s.d. *$P$ value $< 0.05$, according to Mann–Whitney test. (**g**) Double DNA/RNA FISH experiments. *flam* DNA (green) and *flam* transcripts (red) are respectively detected with DNA probe and 508 riboprobe. DNA is stained with Hoescht (white).

and are likely visualized during their transfer from the nucleus to the cytoplasm.

**Exportins Nxt1 and Nxf1 are involved in *flam* RNA export**. To identify the proteins required for the transport of *flam* precursors to Yb-bodies, we looked for factors involved in the nuclear export of *flam* RNAs. Three genome-wide screens for identifying factors implicated in the piRNA pathway either in the germ line or in the somatic follicular cells have been performed[21–23]. Interestingly, knockdown of the central RNA export factors Nxf1 and Nxt1, which are involved in general mRNA export also prevents piRNA production and leads to TE derepression[24,27,28]. *Nxt1* ovarian somatic knockdown (*SKD*) using the *traffic-jam (tj)*-Gal4 driver leads to female sterility and atrophic ovaries, which can however be dissected for analysis. The level of *piwi* and *yb* transcripts was examined in the *SKD* lines. No depletion was detected by quantitative PCR with reverse transcription (RT–qPCR) for both genes. Nevertheless, we found that depletion of *Nxt1* and *Nxf1* in ovarian somatic cells causes loss of Piwi nuclear localization and leads to TE derepression (Supplementary Fig. 1A,B)[22]. The decrease in *Nxt1* and *Nxf1* RNA level was analysed by RT–qPCR on whole ovaries. Because *Nxt1* and *Nxf1* are expressed both in the germline and in ovarian somatic cells, any decrease driven only in the follicle cells can be hardly detected (Supplementary Fig. 1C), even though the mutations cause defects in the piRNA pathway such as loss of Piwi nuclear localization and TE derepression (Supplementary Fig. 1A,B).

*Nxt1-* and *Nxf1-SKD* mutants show striking differences in *flam* precursor localization compared with WT flies. We observed a clear increase in the proportion of follicle cells with a nuclear accumulation of *flam* transcripts in *Nxt1-* and *Nxf1-SKD* (53%, $n = 1,527$ and 27%, $n = 1,005$, respectively) compared with WT *ISO1A* (15%, $n = 1,764$) (Fig. 2a) correlated with a decrease in the proportion of cells with cytoplasmic accumulation (15% and 20%, respectively, versus 39% in WT). Moreover, the proportion of follicle cells with *flam* transcripts detected on the lamin signal was lower in *Nxt1-SKD* flies than in WT (respectively 31% and 46%) but not in *Nxf1-SKD* mutants (52%).

In addition, there was an overall increase in *flam* nuclear staining detected by larger foci in *Nxt1-SKD*, suggesting a nuclear retention of *flam* precursor in this exportin mutant (Fig. 2b and Supplementary Fig. 2). RT–qPCR experiments indicated that the level of *flam* transcripts in *Nxt1-SKD* ovaries was higher than in WT ovaries probably because *flam* precursors are not exported to the cytoplasm to be processed (Fig. 2c). Taken together, these data show that the Nxt1–Nxf1 complex is involved in the export of *flam* transcripts.

If *flam* precursors are not exported to the cytoplasm in *Nxt1* or *Nxf1-SKD* mutants, we should expect that *flam* piRNA precursors cannot be processed into piRNAs. Thus, we sequenced small RNAs from mutant fly-ovaries and examined the level of piRNAs mapping to *flam*. We observed a large decrease in the levels of *flam* piRNAs in *Nxt1-SKD* flies compared with WT flies (5 versus 41 rpkm) (Fig. 2d). As expected, this decrease was observed all along the *flam* locus (Fig. 2e). The decrease in the level of *flam* piRNAs was low in *Nxf1-SKD* flies (37 rpkm) with no significant decrease along the locus (Fig. 2d,e).

We previously reported that *flam* transcripts are channelled within the nucleus from their site of transcription to the distant Dot COM, thus we decided to investigate whether *Nxt1-* and *Nxf1-SKD* also affect this intra-nuclear traffic of piRNA precursors. Double DNA/RNA FISH experiments were performed using *flam 508* RNA probe and a DNA probe made from the *DIP1* gene, a unique genomic sequence flanking the *flam* locus. As previously reported, two distinct foci were detected in

93% of the 346 WT *ISO1A* nuclei examined due to the accumulation of *flam* transcripts away from the genomic *flam* piRNA cluster. Surprisingly, in *Nxt1-* and *Nxf1-SKD* flies, *flam* RNA and DNA co-localized in 43% and 38% of 438 and 532 nuclei examined, respectively (Fig. 2f,g and Supplementary Fig. 3).

Thus, the Nxt1–Nxf1 complex is required for the efficient transfer of *flam* precursors away from their site of transcription.

**EJC is required to transfer *flam* transcripts to Dot COM**. Exportins Nxf1 and Nxt1 are transiently interacting factors of EJC that associates with transcripts concomitantly to splicing[30]. Together with Nxf1 and Nxt1 proteins, the two core EJC factors Mago and Tsu and the EJC accessory proteins RnpS1 and Acn have been involved in the piRNA pathway biology, with their mutation leading to TE derepression[21,23,35–37]. EJC works as a platform linking RNA splicing and export, and facilitates the recruitment of Nxf1 to spliced mRNAs[30]. We investigated whether components of EJC could, like the Nxt1–Nxf1 complex, play a role in the export and/or intra-nuclear traffic of *flam* transcripts. We also tested UAP56 protein, a putative RNA helicase known to transiently interact with EJC that is required for spliceosome assembly[23,38,39]. The sub-cellular localization of *flam* transcripts was assessed in *SKD* flies depleted for *Mago*, *RnpS1*, *Tsu*, *Acn* and *UAP56*.

In preliminary experiments, we performed four tests. First, we tested Piwi localization in the depleted lines. We found that Piwi is absent from the nucleus of these *SKD* lines with two exceptions, *Mago-* and *UAP56-SKD* flies, which have a nuclear Piwi signal either faint or comparable to that of WT flies, respectively (Fig. 3a). Second, we examined the level of the corresponding knocked-down transcripts in the *SKD* lines. As reported above, because these genes are also expressed in the germline, a depletion was only observed in *Tsu-* and *UAP56-SKD* lines, while depletion was hardly detected in *Mago-* and *Rnps1-SKD* and not in *Acn-SKD* (Supplementary Fig. 4A). These variations can potentially be attributed to differences in the expression of these genes within the germline. Thus, we further verified that the silencing exerted on somatic TEs was lost in all these mutants (Supplementary Fig. 4B). Third, since splicing of some mRNAs is impeded in EJC mutants[36,37], we performed RT–PCR using primers in *flam* exon 1 and exon 2 to verify whether *flam* splicing is affected. We found that *flam* intron 1 is correctly spliced in all these *SKD* lines (Fig. 3b,c). Fourth, we analysed by RT–qPCR the impact of *Mago-* and *RnpS1-SKD* on the expression of major genes involved in the somatic piRNA pathway like *piwi*, *armitage (armi)*, *maelstrom (mael)* or *yb* (Supplementary Fig. 5). We found that the expression of none of these genes is affected in these mutants except *piwi* whose expression decreases in *Rnps1*-depleted line as earlier reported[36,37] (see Discussion).

Immuno-FISH experiments were then performed to examine *flam* export in *SKD* mutants for EJC components and UAP56. As seen in Fig. 3d and Supplementary Fig. 6, *flam* transcripts accumulate in a single focus in all depleted lines. In addition, the proportion of cells having cytoplasmic, transmembrane and nuclear accumulation of *flam* transcripts in mutant lines was similar in *SKD* lines and WT (Fig. 3e). Overall, these data indicate that in EJC and *UAP56*-depleted lines, *flam* transcripts are properly spliced and exported to the cytoplasm.

To assess piRNA production, we sequenced ovarian small RNAs from *Mago-* and *RnpS1-SKD* lines. piRNAs mapping to *flam* were detected at a similar level in *Mago-SKD* and WT flies (Fig. 4a,b). Thus, although somatic TE silencing is altered in *Mago*-depleted flies, *flam* precursors nevertheless retain the ability to be processed into piRNAs. Surprisingly, a distinct

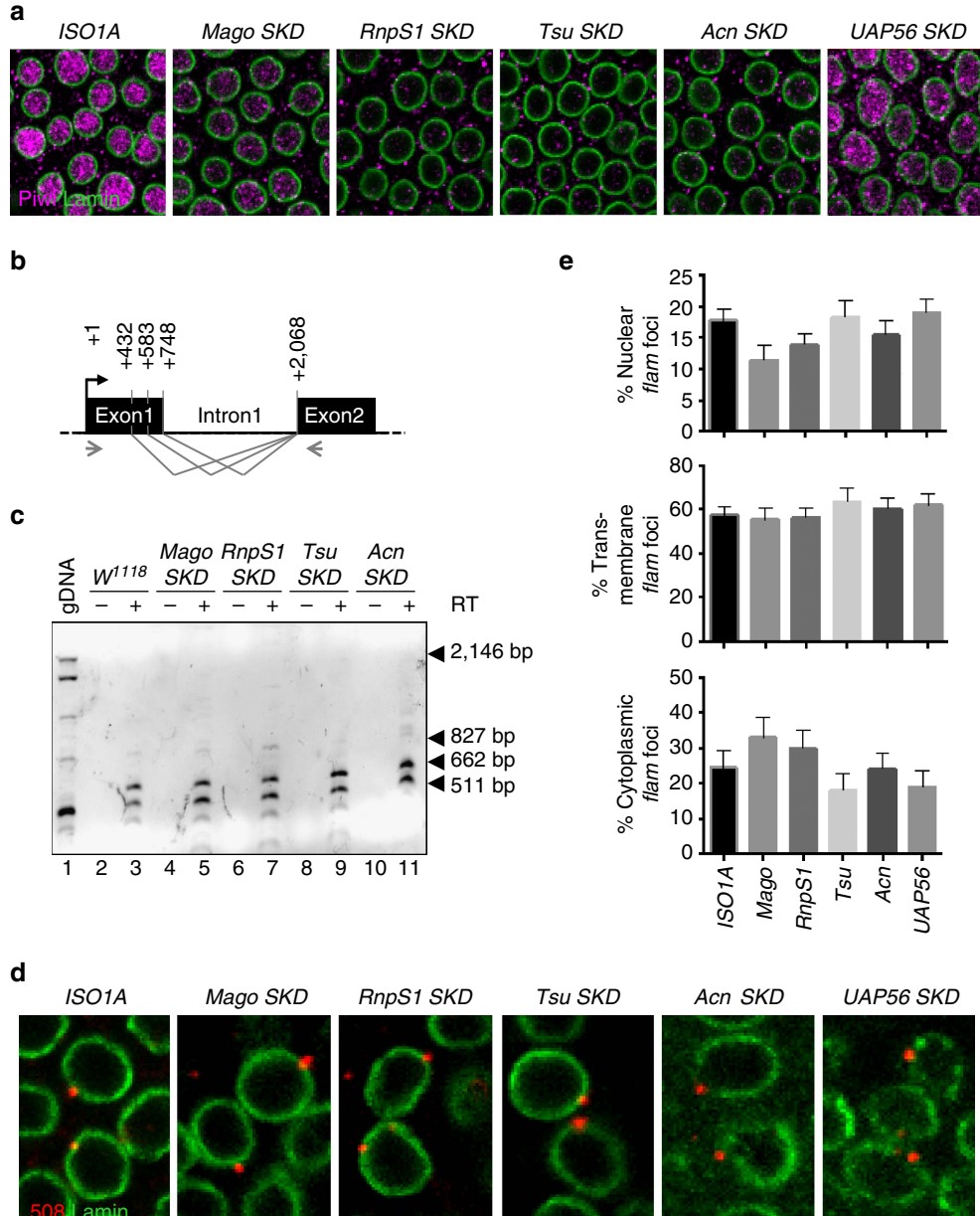

**Figure 3 | *flam* transcripts are correctly spliced and exported in *EJC*- and *UAP56*-depleted lines.** (**a**) Piwi (magenta) and nuclear membrane (green) are visualized by immunofluorescence using anti-Piwi and anti-lamin antibodies, respectively, in *ISO1A* and *Mago*-, *RnpS1*-, *Tsu*-, *Acn*- and *UAP56-SKD* lines. (**b**) Genomic structure of *flam* piRNA cluster 5′ end showing donor (position +432, +583, +748) and acceptor (position +2,068) splicing sites regarding +1 position of *flam* TSS (black arrow). Alternative splicing of intron 1 is shown above by the grey lines. Grey arrows indicate the position of primers used for RT–PCR whose sequence is given in Methods section. (**c**) RT–PCR on WT (*w1118*) genomic DNA and cDNAs from *w1118*-, *Mago*-, *RnpS1*-, *Tsu*- and *Acn-SKD* ovaries are obtained after DNAse I treatment of total ovarian extracted RNAs followed (+RT) or not (−RT) by reverse transcription. Amplification of unspliced molecule gives rise to 2,146 bp DNA (lane 1). Spliced RNAs resulting from first, second or third donor splicing sites give rise to 827, 662, 511 bp fragments (lanes 3, 5, 7, 9, 11). Black arrowheads indicate the position of specific amplified DNA. (**d**) *flam* transcripts (red) and nuclear membrane (green) are visualized by RNA-FISH coupled to immunofluorescence using *flam* 508 RNA probe and anti-lamin antibody, respectively, in ovarian follicle cells of WT *ISO1A* and *Mago*-, *RnpS1*-, *Tsu*-, *Acn*- and *UAP56-SKD* lines. (**e**) Quantitative analysis of *flam* transcript localization in ovarian follicle cells of these depleted lines. The percentage of cells with nuclear, transmembrane or cytoplasmic foci is plotted. Error bars represent s.e.m. $n = 13$ (*ISO1A*), $n = 13$ (*Mago*), $n = 14$ (*Rnps1*), $n = 12$ (*Tsu*), $n = 16$ (*Acn*), $n = 12$ (*UAP56*), $n$ is the number of independent experiments.

reduction in the production of *flam* piRNAs was observed in *RnpS1*-depleted flies (Fig. 4a,b). Since the export of piRNA precursors appeared to be unaffected (Fig. 3d,e), this suggests that an additional step for the production/stability of piRNAs is missing.

To examine the intra-nuclear traffic of *flam* piRNA precursors, the relative localization of nuclear *flam* transcripts to *flam* genomic locus was analysed by double DNA/RNA FISH. Strikingly, an increase in the co-localization of *flam* precursor with its genomic site of transcription was observed (Fig. 4c,d and Supplementary Fig. 7). In *Mago*- and *RnpS1*-depleted flies, DNA and RNA co-localize in 40% ($n = 553$) and 38% ($n = 533$) of follicle cells, respectively, while *flam* transcripts are detected at their site of transcription in only 7% ($n = 346$) of WT cells.

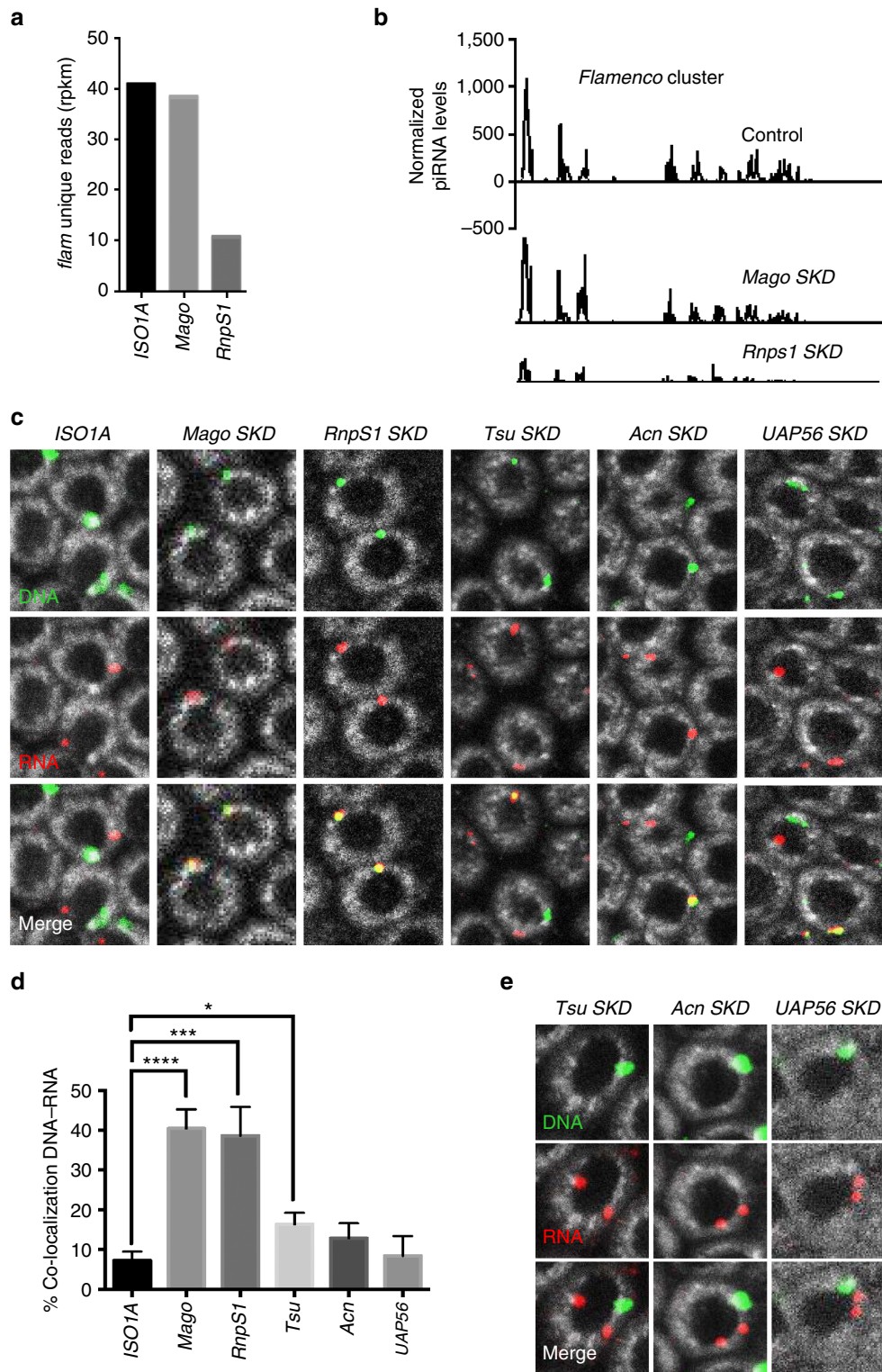

**Figure 4 | Components of EJC and UAP56 protein are required for *flam* transcript intra-nuclear traffic to Dot COM.** (**a,b**) Changes in steady-state levels of *flam*-unique 23–29 nt piRNAs in ovarian cells of *ISO1A* and *Mago*- and *RnpS1-SKD* lines measured by small RNA sequencing (normalized to one million genome-mappable reads). (**a**) Density plot of unique 23–29 nt piRNAs mapping to the *flam* piRNA cluster. The *Y* and *X* axes in **b** are identical for the three graphs. *X* axis represents the *flam* 180 kb starting from its transcription start site on the left. *Y* axis represents the quantity of piRNAs normalized to one million genome mappable reads. (**c,e**) Double DNA/RNA FISH experiments staining *flam* DNA (green) and *flam* transcripts (red) in ovarian follicle cells of WT *ISO1A* and *Mago*-, *RnpS1*-, *Tsu*-, *Acn*- and *UAP56-SKD* lines (**c**) or of *Tsu*-, *Acn*- and *UAP56*-depleted lines (**e**). DNA is stained with Hoescht (white). (**d**) Quantitative analysis of relative localization between *flam* DNA and *flam* transcripts in ovarian follicle cells. The percentage of cells with a DNA–RNA co-localization is plotted. Error bars represent s.e.m. $n = 32$ (*ISO1A*), $n = 40$ (*Mago*), $n = 26$ (*Rnps1*), $n = 34$ (*Tsu*), $n = 34$ (*Acn*), $n = 18$ (*UAP56*), *n* is the number of independent experiments. ****$P$ value $< 0.0001$, ***$P$ value $< 0.001$, *$P$ value $< 0.05$, according to Mann–Whitney test.

To a lesser extent 16% ($n = 430$), of follicle cells have a co-localization between *flam* RNA and DNA in *Tsu*-depleted line (Fig. 4d). No significant difference was found in *Acn*- and *UAP56*-depleted lines compared with WT. However, in all EJC-depleted lines, including *Acn* and *UAP56* mutant lines, the transfer of *flam* transcripts to Dot COM appears to be affected. Indeed, in a few cells, we also observed two distinct Dot COMs, neither of which co-localized with *flam* DNA (Fig. 4e and Supplementary Fig. 8), suggesting that gathering of *flam* transcripts during the transfer from their site of transcription to a single focus of accumulation is affected in all these mutant lines.

Altogether, these results indicate that EJC components are required to transfer *flam* transcripts from their site of transcription to the distant nuclear Dot COM site. They also show that *flam* transcripts reach a single site of accumulation.

**Export of *flam* precursors is required for Yb-body formation.** Nuclear Dot COM is often juxtaposed with the cytoplasmic Yb-body[17]. In *Nxt1-SKD* mutants, *flam* transcripts are neither shuttled from their site of transcription to Dot COM nor exported to the cytoplasm. To a lesser extent, the transfer of *flam* precursors to Dot COM is also affected in *Nxf1-SKD*. We sought to examine the position of the Yb-bodies in these two *SKD* lines. Immunofluorescence experiments were performed using anti-Armi and anti-Yb antibodies, which are two major constituents of Yb-bodies. Strikingly, very few Y-bodies were visualized in *Nxt1-SKD* mutants. Only 26% ($n = 1,333$) of follicle cells, still harboured Yb-bodies compared with 63% ($n = 1,401$) in WT flies. Instead, Armi and Yb proteins were dispersed throughout the cytoplasm of follicular cells giving rise to a diffuse cytoplasmic signal (Fig. 5a,b and Supplementary Fig. 9A). In *Nxf1-SKD* flies, a faint Armi cytoplasmic staining was observed. Yb-bodies were also detected but they looked smaller than in WT. These data suggest that either export or intra-nuclear traffic of *flam* precursors could be required for Yb-body assembly.

In knockdown lines for EJC components, the intra-nuclear traffic of *flam* is affected but not its export. We first investigated whether Yb-body formation is altered in *Mago*- and *RnpS1-SKD* lines. Follicle cells having at least one Yb-body were found in a similar proportion to that in WT flies (Fig. 5c,d and Supplementary Fig. 9B). No visible cytoplasmic diffuse Armi or Yb signal was registered. A correct assembly of Yb-bodies has been previously reported in other EJC mutants[37]. These results indicate that it is the export of *flam* transcripts and not their intra-nuclear traffic that is the signal for Yb-body assembly.

Immuno-FISH experiments were then used to examine the position of Yb-bodies in *Mago*-, *RnpS1*-, *Tsu*-, *Acn*- and *UAP56*-SKD lines. They showed that cytoplasmic Yb-bodies are juxtaposed to nuclear *flam* transcripts (Fig. 6a,b and Supplementary Fig. 10). Since the nuclear traffic of *flam* transcripts is disrupted in these mutants, it is expected that Yb-bodies are then juxtaposed to *flam* transcription site. By combining DNA FISH and immunofluorescence, we confirmed that a higher proportion of Yb-bodies is juxtaposed to the *flam* genomic locus in *RnpS1-SKD* line compared with WT (Fig. 6c,d). Interestingly, in a few cells displaying two Dot COMs, two Yb-bodies were assembled, each of them facing one Dot COM (Fig. 6e and Supplementary Fig. 11). Taken together, these data indicate that Yb-bodies assemble at the site where accumulated *flam* precursors are exported irrespective of whether transcripts are channelled away or not from their transcription site. They further indicate that *flam* precursor traffic within the nucleus occurs in a process that is upstream of Yb-body formation.

*flam* is the major piRNA cluster expressed in ovarian somatic cells. Other piRNA clusters are expressed in these cells but they

weakly participate in the production of piRNAs[1]. We investigated whether transcripts originating from other piRNA clusters are sufficient to induce the assembly of Yb-bodies. We used the *flam BG02658* allele (*flamBG*), which, owing to a P-transgene inserted upstream of *flam* transcription start site, produces no *flam* transcripts[40]. Homozygous mutant ovaries are small and atrophic with very few egg chambers. However, we were able to examine a few egg chambers in which an epithelium of follicle cells was correctly formed. Mutant follicle cells have smaller, fainter and many more Armi foci than do WT cells. In addition, we observed a diffuse cytoplasmic staining in homozygous *flamBG*, similar to what was seen in *Nxt1-SKD* follicle cells, suggesting that a fraction of Armi protein fails to accumulate in foci (Fig. 6f). These data indicate that *flam* piRNA cluster acts as a master locus in ovarian follicle cells for Yb-body assembly.

## Discussion

In this work, we examined how the piRNA precursors are transferred from their site of transcription to the cytoplasmic structure where they are processed into primary piRNAs. We report that, in the follicle cells of *Drosophila* ovaries, EJC components and their associated proteins Nxf1, Nxt1 and UAP56 play a crucial role in this process that displays three steps: recognition of the precursor, transfer through the nucleoplasm away from the DNA locus and export to the cytoplasm. We further show that export of *flam* transcripts is required for the presumed processing structure called the Yb-body to assemble in the cytoplasm.

We found that the traffic of *flam* piRNA precursors within the nucleus from the DNA locus to a distal nuclear structure previously called Dot COM depends on EJC factors Mago, RnpS1 and Tsu and on exportins Nxt1 and Nxf1. These data argue that binding of EJC and export proteins occurs shortly before or after release of *flam* precursors from their site of transcription and provide a link between *flam* precursor splicing and its downstream nuclear transfer to Dot COM. At present, we do not know whether interaction between the *flam* precursor and these factors is direct or indirect. EJC components may each bind *flam* precursor directly and independently. Alternatively, some may bind directly, whereas others may only associate indirectly through interactions within the complex. It is also possible that these interactions occur with other yet unknown factors. Moreover, we cannot exclude that through a more general effect on mRNA processing and transport, EJC mutants may indirectly affect *flam* export and the piRNA pathway. However, it has been verified that depletion of each of these EJC components had no significant impact on the level of transcripts of genes involved in the somatic piRNA pathway except for *piwi* whose transcripts level decreases in *RnpS1-SKD* (Supplementary Fig. 5A and ref. 36).

Nxf1 and Nxt1 are involved in *flam* export (Fig. 2a). However, depletion of *Nxf1* has a limited impact on piRNA production. It is possible that the redundancy which has been reported between export components binding nuclear pore complex may compensate depletion of *Nxf1* (refs 41,42). Alternatively, in our *SKD* mutant, a residual Nxf1 function could be sufficient to allow the export of enough *flam* precursors that are processed to an almost WT amount of piRNAs (37 and 41 rpkm, respectively) (Fig. 2e). Nxt1 plays an essential role in the export of piRNA precursors in follicle cells. Several studies have reported the ability of Nxt1 to mediate nuclear mRNA export by controlling the interaction of Nxf1 with components of the nuclear pore[29,43]. Our data further extends the critical importance of Nxt1 to this non-coding RNA export.

Components of the EJC assume different functions to connect nuclear pre-mRNA maturation to mRNA fate. Their individual

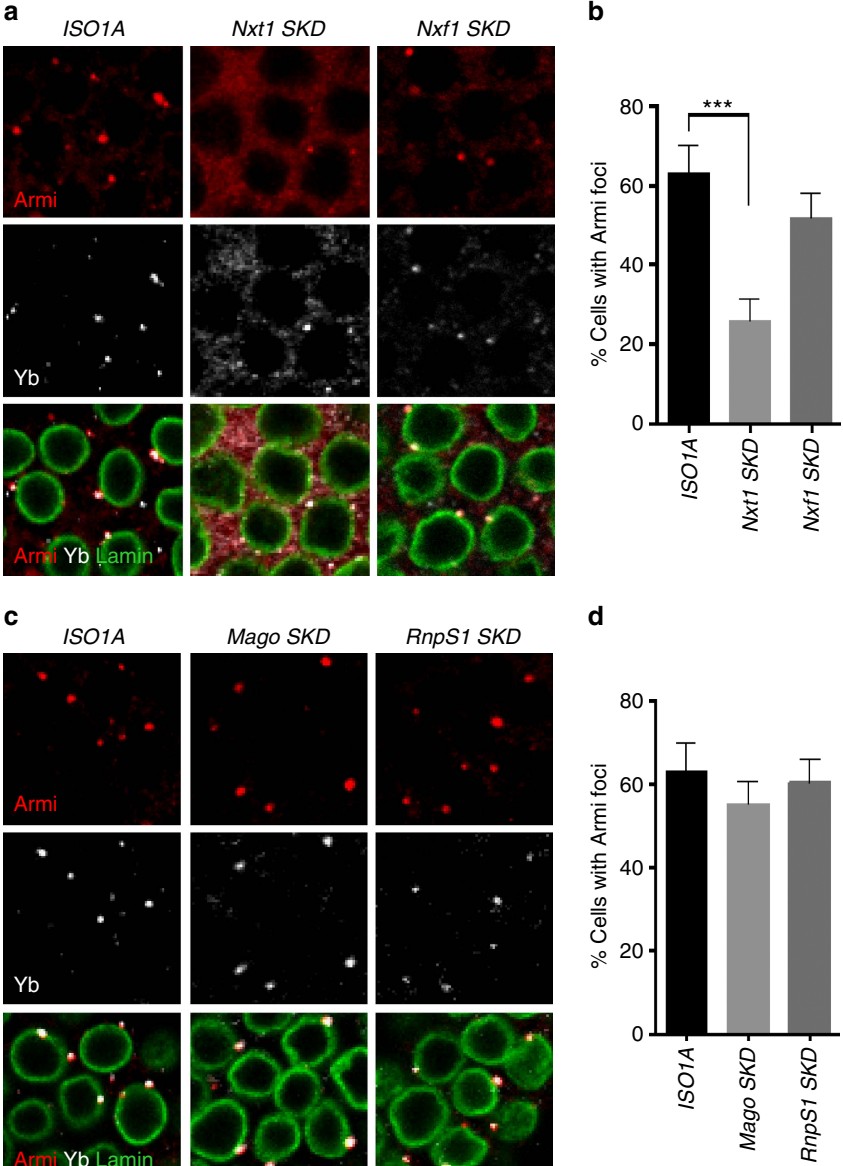

**Figure 5 | Yb-body formation is altered in *Nxt1*- and *Nxf1*- but not in *EJC*-depleted follicle cells.** (**a**,**c**) Armi (red), Yb (white) and nuclear membrane (green) are visualized by immunofluorescence in *ISO1A* and *Nxt1*- and *Nxf1-SKD* follicle cells (**a**), and in *ISO1A* and *Mago*- and *RnpS1*-depleted cells (**c**). (**b**,**d**) Quantitative analysis of Armi foci in *ISO1A* (n = 16), *Nxt1*-(n = 19) and *Nxf1-SKD* (n = 16) follicle cells (**b**) and in *ISO1A* (n = 16) and *Mago*-(n = 15) and *RnpS1*-depleted cells (n = 15), n is the number of independent experiments (**d**). The percentage of cells harbouring one or more Armi foci is plotted. Error bars represent s.e.m., ***P value < 0.001, according to Mann–Whitney test.

functions certainly affect different steps in the *flam* transfer to the cytoplasm. Mago and Tsu are known to form a heterodimer required for a tight binding of EJC to the targeted RNAs[30]. RnpS1 and Acn have been implicated in mRNA quality control, pre-mRNA splicing and transcriptional regulation[44]. UAP56 is involved in splicing and mRNA export[45]. However, we could hardly attribute specific roles to these EJC components in the nuclear traffic of *flam*. Their mutation does affect neither *flam* transcription, nor splicing, nor its nucleo-cytoplasmic export. The only step that is affected in all mutants is the intra-nuclear traffic of *flam* transcripts and/or, occasionally, their gathering to a single focus of accumulation (Fig. 4e and Supplementary Fig. 8). Differences depending on the depleted gene were nonetheless observed. First the intra-nuclear traffic of *flam* from the site of transcription to Dot COM is greatly affected in *Mago*- and *Rnsp1-SKD* and to a less extend in *Tsu* mutant. Instead, *Acn*-, *UAP56*-,

as well as *Tsu-SKD* appear to affect the gathering of *flam* precursors into a single focus. Second, *flam* processing into piRNAs is highly decreased in *RnpS1*- or *Tsu*-depleted flies but not in *Mago-SKD* (Fig. 4a,b and ref. 22) or *UAP56* mutants[38]. To better understand the interactions and the role played by each of these factors on *flam* precursor processing, it will be important to find the temporal sequence of binding, if binding to RNA is direct and if some of the components bind the potential piRNA-trigger sequence that is crucial for piRNA production[19].

EJC mutants have been reported to affect mRNA splicing. Notably, excision of the *piwi* fourth intron is impaired and the level of Piwi protein reduced. The decrease in piRNA level observed in *RnpS1* mutant could be due to the loss of Piwi protein. In this case, processed piRNAs would not be able to be loaded onto Piwi and would then be degraded. Why it is not

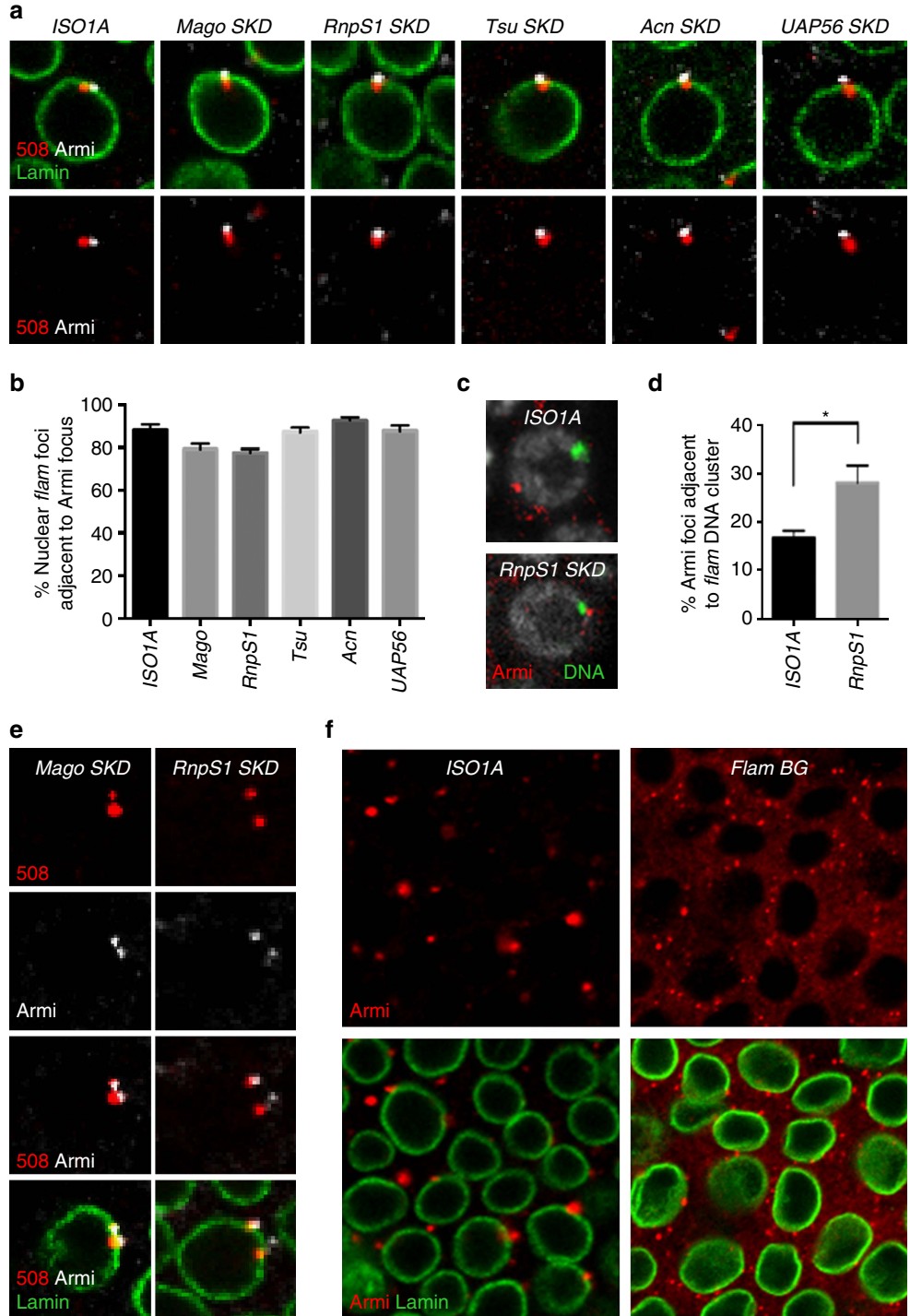

**Figure 6 | Yb-bodies are assembled at the site of export of *flam* transcripts and fail to form foci in absence of *flam* transcripts** (**a**,**e**) *flam* transcripts (red), Armi foci (white) and nuclear membrane (green) are visualized by RNA-FISH coupled to immunofluorescence in ovarian follicle cells of *ISO1A* and *Mago*-, *RnpS1*-, *Tsu*-, *Acn*- and *UAP56-SKD* lines (**a**) or of *Mago*- and *RnpS1-SKD* lines (**e**). (**b**) Quantitative analysis of *flam* transcripts adjacent to Yb-bodies in ovarian follicle cells of these depleted lines. The percentage of cells with a nuclear or trans-membrane RNA adjacent to Armi focus is plotted. Error bars represent s.e.m. $n = 22$ (*ISO1A*), $n = 20$ (*Mago*), $n = 20$ (*RnpS1*), $n = 19$ (*Tsu*), $n = 18$ (*Acn*), $n = 10$ (*UAP56*), $n$ is the number of independent experiments. (**c**) DNA FISH coupled to immunofluorescence labels *flam* DNA (green) and Armi protein in ovarian follicle cells of *ISO1A* and *RnpS1-SKD* lines. (**d**) Percentage of cells with Yb-bodies adjacent to *flam* DNA in ovarian follicle cells of *RnpS1*-depleted line compared with WT. Error bars represent s.e.m. $n = 20$ (*ISO1A*), $n = 24$ (*RnpS1*), $n$ is the number of independent experiments. *$P$ value $< 0.05$, according to Mann–Whitney test. (**f**) Yb-bodies detected using anti-Armi antibody (red) are compared in *ISO1A* and homozygous *flamBG* flies. Anti-lamin antibody stains the nuclear periphery (green).

observed in other EJC mutants is not known but in *Mago* and *UAP56* mutants *Piwi* transcription and splicing is poorly affected[36–38] and Piwi protein is detectable within the nucleus (Fig. 3a). This small amount of WT Piwi could be sufficient to stabilize piRNAs. These data raise an important question. Why are *Mago*, *Acn* and *Tsu* mutants unable to silence somatic

TEs if piRNAs are normally produced? Several hypotheses can be proposed. First, unknown proteins whose mRNA splicing is also affected by the mutations could be required to produce functional piRNAs or to silenced TE at the transcriptional level. Second, the traffic of *flam* precursors away from their transcription site before leaving the nucleus may be required for the piRNAs to become functional. This hypothesis is consistent with the known role of EJC proteins to mark RNAs for downstream stability; However it is still an open question whether the defect in TE silencing observed in *EJC SKD* is directly or indirectly connected to intranuclear trafficking of *flam*.

Unlike somatic cells, germline has evolved a mechanism that inhibits splicing of the nascent piRNA transcript[39]. piRNA clusters expressed in the germline are preferentially dual-strand clusters based on transcription orientation occurring from both strands. The RDC complex composed of Rhino, Deadlock, Cutoff is required for the non-canonical transcription of these dual-strand piRNA clusters and involves 5′ end protection of nascent RNAs and suppression of transcription termination. It was speculated that stalled splicing due to RDC and UAP56 differentiates these piRNA precursors from mRNAs[39,46,47]. Then, our study underlines that the non-canonical transcription of dual-strand piRNA clusters in the germline and the canonical transcription of uni-strand clusters in the follicle cells engage different processes both leading to piRNA biogenesis.

Overall, it can be anticipated that components of EJC and exportin pathways that are well conserved throughout eukaryotes fully participate in the defence system protecting genomes from TE invasions.

When *flam* precursors are channelled within the nucleus to Dot COM, Yb-bodies are detected facing Dot COM. If this intra-nuclear traffic is blocked as in EJC mutants, in which *flam* precursors are exported into the cytoplasm close to their site of transcription, then Yb-bodies are juxtaposed to the *flam* genomic locus. Interestingly, when two *flam* foci are observed in EJC mutants, each of them face one Dot COM, confirming that wherever *flam* transcripts are exported, a Yb-body is always detected in close proximity. Moreover, when *flam* piRNA cluster is not transcribed in flies carrying the *flam*BG allele, Yb-body formation is greatly affected. A clear dispersed Armi cytoplasmic staining suggests that a large part of Armi protein fails to accumulate in one spot. *flam* is a very specific piRNA cluster as it produces the largest amount of piRNAs in follicle cells. We here provide evidence that it also acts as a master locus for Yb-body assembly.

Interestingly, in the *Drosophila* germline, an extrinsic control of nuage assembly and function exerted by nuclear components has also been reported[38]. Like Yb-bodies in the somatic lineage, nuage is the cytoplasmic structure in the germline where piRNA biogenesis is likely to occur. Discrete nuclear foci corresponding to germline piRNA cluster transcripts bound by Rhino and UAP56 proteins have been observed juxtaposed to the cytoplasmic nuage. When mutations affect UAP56, the nuage assembly is disrupted, revealing an upstream requirement of UAP56. Zhang *et al.* proposed that Rhino and UAP56 are connected to the nuage via the nuclear pore complex that could therefore trigger release of cluster transcripts in the nuage. A nuclear envelope-spanning machinery would then be required for piRNA biogenesis. Like in the germline, we found that Yb-body assembly requires *flam* export as an extrinsic control coming from the nucleus. However, as opposed to the germline, UAP56 is not required for Yb-body assembly since its mutants do not disrupt Yb-bodies (Fig. 6a). It is possible that Nxt1 plays a key role in this process by stimulating interactions of exportins with components of the nuclear pore. Further insight into the machinery spanning the nuclear envelope is necessary to enlighten the whole mechanism that will result in the precursor export through the nuclear membrane.

## Methods

**Drosophila stocks.** All flies were kept at 20 °C. Strains *ISO1A*, *W1118*, *BG/FM4* and *traffic jam (tj)*-GAL4/CyO came from the collection of the GReD. *nxt1* [VDRC GD49602] strain was obtained from Vienna *Drosophila* Resource Center. *nxf1/sbr* [34945], *mago* [35453], *rnpS1* [36580], *tsu* [36585], *acn* [53676] and *uap56/hel25E* [33666] strains were obtained from the Bloomington *Drosophila* Stock Center.

**In situ hybridization.** The DNA fragment to prepare the specific *flam* 508 probe to detect *flam* transcripts was PCR amplified from the *ISO1A* line using primers 5′-ATTCTCCTTTCTCAGGATGC-3′ and 5′-GCATTGCTACCTTACGTTTC-3′ and cloned into pGEMT easy vector.

Riboprobe was synthesized by digestion of pGEMT easy plasmids with NcoI or SpeI enzyme, followed by *in vitro* transcription using Sp6 or T7 polymerase and digoxygenin labelled UTP (Roche), DNAse I treatment and purification.

DNA probe are made of eleven PCR amplifications (Supplementary Table 1) labelled with digoxigenin using nick translation kit (Roche).

RNA FISH was performed on ovaries from 2- to 4-days-old flies dissected in 0.2% Tween 20, phosphate-buffered saline (PBT). Ovaries were fixed with 4% formaldehyde/PBT at room temperature (RT) for 30 min, rinsed three times with PBT, post-fixed 10 min in 4% formaldehyde/PBT and washed in PBT. After permeabilization 1 h in PBS-0.3% Triton, prehybridization was performed as follow: 10 min HYB- (Formamide 50%, SSC 5 ×, Tween 0.02%)/PBT 1:1, 10 min HYB-, 1 h HYB+ (HYB- with yeast tRNA 0.5 mg ml$^{-1}$, heparin 0.25 mg ml$^{-1}$) at 60 °C. Ovaries were hybridized overnight at 60 °C with 1 μg riboprobe. Ovaries were then rinsed 20 min in HYB- and in HYB-/PBT 1:1 at 60 °C then four times in PBT at RT before blocking 1 h at RT in TNB 0.3% triton (Perkin-Elmer TSA kit) and immunodetection 1 h 30 min at RT with anti-digoxigenin-HRP (11 207 733 910, Roche, 1:200 dilution) in TNB 0.2% tween. Ovaries were rinsed three times in PBT, incubated 10 min with TSA-Cy3 in amplification diluent (Perkin-Elmer, 1:50 dilution), and rinsed.

When coupled to immunofluorescence, RNA staining was followed by incubation with mouse anti-lamin antibody (ADL67-10, Hybridoma, 1:300 dilution) and goat anti-Armi antibody (sc-34564, Santa Cruz, 1:250 dilution). Secondary antibodies coupled to Cy3 or Alexa-488 were used.

For DNA/RNA *in situ* hybridization, RNA staining was followed by treatment with 200 μg ml$^{-1}$ RNase A for 2 h after which the ovaries were transferred to FISH hybridization buffer containing 50% formamide, 4× SSC, 0.1% Tween 20, 0.1 M NaH$_2$PO$_4$. DNA was denatured for 15 min at 80 °C and hybridization with DNA probe was carried out O/N at 37 °C. After washes, the ovaries were first incubated with glycine 0.1 M–0.1% Tween HCL pH 2.2 before washes and DNA staining. DNA staining was performed by blocking for 1 h at RT in TNB, immuno-detection for 1 h 30 min at RT with anti-Digoxigenin-HRP (11 207 733 910, Roche, 1:200 dilution) in TNB 0.3% Triton, washes three times in PBT, incubation for 10 min with TSA-Cy3 in amplification diluent (Perkin-Elmer, 1:25 dilution), rinsing three times in PBT and staining with Hoescht.

For DNA FISH coupled to immunofluorescence, ovaries from 2- to 4-day-old flies were dissected in PBT (PBS-0.2% Tween) fixed with 4% formaldehyde/PBT at RT for 30 min, rinsed three times with PBT, post fixed for 10 min with 4% formaldehyde/PBT at RT and rinsed in PBT. The ovaries were then permeabilized for 1 h in PBS-0.3% Triton, rinsed three times with PBT and incubated for 1 h in TNB (Perkin-Elmer TSA kit) 0.3% Triton prior to staining with mouse anti-Armi antibody (kindly provided by M. Siomi). Incubation with secondary antibodies coupled to Cy3, Cy-5 or Alexa-488 was followed by treatment with 200 μg ml$^{-1}$ RNase A for 2 h. The ovaries were then transferred to FISH hybridization buffer containing 50% formamide, 4× SSC, 0.1% Tween 20, 0.1 M NaH$_2$PO$_4$. DNA was denatured for 15 min at 80 °C and hybridization with DNA probe was carried out O/N at 37 °C. After washes, the ovaries were treated for DNA staining.

**Immunofluorescence.** Ovaries from 2- to 4-day-old flies were dissected in PBT (PBS-0.2% Tween) fixed with 4% formaldehyde/PBT at RT for 20 min, rinsed three times with PBT, incubated for 1 h in PBS-0.3% Triton, rinsed three times with PBT and incubated for 1 h in TNB (Perkin-Elmer TSA kit) 0.3% Triton prior to staining with mouse anti-lamin antibody (ADL67-10, Hybridoma, 1:300 dilution), goat anti-Armi antibody (sc-34564, Santa Cruz, 1:250), rabbit anti-Yb antibody (kindly provided by J. Brennecke, 1:1,000 dilution) or rabbit anti-Piwi antibody (sc-98264, Santa Cruz, 1:500 dilution). Secondary antibodies coupled to Cy3, Cy-5 or Alexa-488 were used. Three-dimensional images were acquired on Leica SP5 and Leica SP8 confocal microscopes using a × 40 objective.

**RT-PCR analysis.** Total RNA was isolated from 10 pairs of ovaries from 2- to 4-day-old flies with Trizol (Ambion). Following DNAse I treatment, cDNA was prepared from 1 μg RNA by random priming of total RNA using Superscript IV Reverse Transcriptase (ThermoFisher Scientific).

PCR were performed with Phusion DNA polymerase (New England Biolabs) using e1F 5′-AGTTGCTTTATGACGCCGGGC-3′ and e2R 5′-GCTAGGAAGCTA ATTGATCAATGACAAC-3′ primers.

qPCR were performed using standard techniques. All experiments were conducted in biological triplicates with technical triplicates. Relative RNA levels were normalized to *rp49* levels. Fold enrichments were calculated in comparison with RNA levels obtained from WT *ISO1A* line. Primer sequences for RT–qPCR analyses are listed in Supplementary Table 2.

**Small RNA sequencing.** Total RNA was isolated from 10 pairs of ovaries from 2- to 4-day-old flies with Trizol (Ambion). Deep sequencing of 18–30 nt small RNAs was performed by Fasteris S.A. (Geneva/CH) on an Illumina Hi-Seq 2500. Illumina reads were matched to release 6 of the *Drosophila* genome. Only reads mapping uniquely to the genome were used for further analysis. All libraries were normalized based on the number of mapped reads. Size profile for *flam* piRNA cluster was obtained by extracting the abundance and read length of 23–29 nt sequences uniquely matching to this locus.

**Data availability.** All Illumina data sets that support the findings of this study have been deposited at GEO with the accession codes: BioProject ID PRJNA348903. The data that support the findings of this study are available from the corresponding author upon reasonable request.

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

## Acknowledgements

This work was supported by grants from the Agence National pour la Recherche (ANR—Plastisipi project). E.B. and A.S. received a grant from the Région Auvergne and Université d'Auvergne, respectively. We are grateful to R. Pogorelcnik and P. Pouchin for

the pipeline used for the small RNA analysis, and J. Brennecke and M. Siomi for antibodies.

## Author contributions

C.D. and A.S. performed experiments; E.B. performed bioinformatic analysis; C.V. supervised the project; C.D., E.B. and C.V. wrote the article.

## Additional information

**Competing financial interests**: The authors declare no competing financial interests.

**How to cite this article**: Dennis, C. *et al.* Export of piRNA precursors by EJC triggers assembly of cytoplasmic Yb-body in *Drosophila*. *Nat. Commun.* **7**, 13739 doi: 10.1038/ncomms13739 (2016).

**Publisher's note**: 

