## [Peer Review File · Nature Communications]

Reviewers' Comments:

Reviewer #1 (Remarks to the Author)

The manuscript by Dennis et al. presents a set of data linking the EJC
complex and its associated exportins Nxt1 and Nxf1 to the nuclear
transfer and export of piRNA precursor transcripts originating from the
flamenco locus in Drosophila follicle cells. The authors show that nuclear
transport of the flam transcript is compromised when EJC components are
knocked down, while depletion of Nxt1 and Nxf1 affects flam RNA export
to the cytoplasm. In addition, they show that assembly of cytoplasmic Yb-
bodies, the piRNA processing machineries in OSCs, is strongly affected in
export-deficient mutants, but not in EJC mutants. Based on this data, the
authors conclude that flam transcripts are directly required for Yb-body
assembly, and that nuclear transport of piRNA precursors is essential for
the production of functional piRNAs.

The manuscript is compactly written and fairly easy to follow. The results
are interesting and of clear importance to the field. The finding that piRNA
precursors hijack the general RNA splicing and transport machineries for
their biogenesis, is also generally interesting, and consistent with their
essential function in genome defense. The data is mostly well presented
and clearly explained, however the following points should be addressed:

- 1. Several figures are shifted or otherwise misaligned (e.g. Fig 2A, C, Fig
3C, etc.), making them difficult to understand!
- 2. It would be worthwhile discussing briefly if and how the general effects
of exportin and EJC component knock-downs on mRNA processing and
transport may affect the results. Eg. is the processing or expression of
other piRNA factor proteins altered in these cell lines?
- 3. Why does the Nxf1-SKD have a much weaker effect than Nxt1-SKD? Is
the reduction of piRNAs (Fig 2D,E), the loss of DNA-RNA co-localization
(Fig 2F), and the reduction of Yb-body assembly (Fig 5B) even significant
for this mutant?
- 4. Please explain more clearly why are the effects of the various EJC
component depletions diverse.
- 5. The discussion contains much repetition from results. While
summarizing the results here is advantageous and should be kept, some
repetition could be removed. Instead, the authors could discuss to what
extent the role of EJC/exportins in piRNA trafficking may be conserved in
germ cells, and/or other organisms. Similarly, do the authors expect that
the role of long piRNA precursors in initiating the assembly of their
processing machineries is generalizable?
- 6. Statistical tests have not been performed throughout the manuscript.
Error bars should be shown and p values should be given whenever
possible.

Minor points:

7. The distinctive roles of EJC and exportins in piRNA precursor trafficking
are not clear from the Abstract.

8. Aub and Armi should be spelled out when first used. *D. melanogaster*
also.

9. Fig 2B: it should be stated somewhere why some cells do not have
flam foci.

10. In Fig 2E (and Fig 4B) are the Y-axes identical for the 3 graphs?
Should be noted in the legend.

11. On page 9, second paragraph, the third sentence starting with "As a
consequence..." is confusing. Does this refer to the RnpS1 knock-down
cells?

12. In Fig 4D and Fig 6D, should the WT bars not be more similar to each
other?

Reviewer #2 (Remarks to the Author)

Manuscript No.: NCOMMS-16-10010-T

"Export of piRNA precursors by EJC triggers assembly of cytoplasmic Yb-
body in *Drosophila*"

Cynthia Dennis, Emilie Brasset, and Chantal Vaury

This manuscript has two major flaws. First, little of the data are reported
with measurements of error and statistical significance. Given the very
large number of wild-type versus mutant comparisons in which the
measured change is considerably less than twofold, such standard
statistical values are essential to evaluating the strength of the data in
supporting the conclusions the authors draw. Second, the authors
conclude that each mutant tested reflects a direct role for the gene
product in intra-nuclear transfer, nuclear export, or Yb-body localization
of piRNA precursors. In truth, genetic and molecular data can rarely
distinguish direct from indirect effects. The data presented here cannot
distinguish between direct and indirect roles for exportins or EJC
components in piRNA biogenesis. I recommend that the manuscript be
rejected.

Major Concerns

(1) In Figure 1, the authors report that the *flamenco* transcript
localizes to three distinct cellular compartments: a body inside the
nucleus, a body coincident with the nuclear periphery, and a body in the
cytoplasm. In contrast, Murota et al. (2014) reported detecting only
cytoplasmic "Flam Bodies." Given the discrepancy, it is unclear whether
the observed localization patterns are an artifact of confocal imaging.
Clearly additional markers for the three compartments are needed.

(2) No consistent metric is used for *flam* RNA localization: some
figures report three classes (inside nucleus, peripheral, or in cytoplasm)

while others divide the *flam* RNA between just two classes or
present data for only a single category. It is impossible to make
thoughtful comparisons among the mutants with such data.

(3) Small and often contradictory changes in the localization of Armi and
*flam* RNA, or in the level of mature piRNA are presented as
meaningful, or are ignored without justification. Much of the apparent
changes between mutants may simply reflect experimental variation.

104 Minor Points

(1) Abstract: There is no evidence that piRNA processing occurs in Yb
bodies; all current data simply identifies the subcellular regions containing
the highest local concentrations of piRNA precursors and processing
components, not where the actual endonucleolytic and exonucleolytic
steps or PIWI-protein loading occur.

(2) Introduction: piRNA precursors in mammals do not come from TE
clusters, even those piRNAs that target transposons. The entire
Introduction describes piRNA biology in flies, but is written as if it applies
more generally to all animals (metazoan).

(3) The Introduction completely ignores recent papers suggesting that the
primary piRNA biogenesis pathway operating in the fly ovarian follicle
cells does not exist in the fly germline (Mohn et al. 2015; Han et al. 2015;
Senti et al. 2015; Wang et al. 2015). It also completely ignores that the
majority of germline piRNA clusters are transcribed in a manner distinct
from the somatic *flam* cluster: dual-strand vs. uni-strand transcription;
non-canonical vs. canonical pol II transcription; unspliced vs. spliced
piRNA precursor (Klattenhoff et al. 2009, Zhang et al. 2014, Mohn et al.
2014).

(4) The Introduction never acknowledges that the involvement of UAP56
in piRNA biogenesis was first reported by Zhang et al. (Ref. 33). Not until
the introductory paragraph of the Results section do we learn that: "We
also tested UAP56 protein, a putative RNA helicase known to transiently
interact with EJC that is required for spliceosome assembly and somatic
TE silencing (22, 33)." However, Ref. 33 shows that UAP56 is
*dispensable* for somatic TE silencing!

(5) Results: It is unusual to refer to one's own data as "accurate," but
even more unusual to refer to it as "quantitative" and then immediately
refer to "a few cells" rather than an actual percentage or quantity.

138 Reviewer #3 (Remarks to the Author)

General comments. These interesting studies regarding *Drosophila*
ovarian somatic cell piRNAs report two overall conclusions:

(1) intranuclear movement of pre-piRNA from the site of transcription to
the DotCOM licences future pi-RNAs for function in TE silencing. The
authors present convincing imaging (DNA/RNA FISH and DNA/RNA/FISH-
IF) documenting that deficiencies in the Nxf1/Nxt1 mRNA nuclear export
machinery result in failure of flam pre-piRNAs to transit from their site of
transcription to intranuclear foci (DotCOM) as well as a failure to be
exported to the cytoplasm. In contrast, knockdowns of EJC components,
while resulting in fewer DotCOM foci, do not affect pre-piRNA splicing,
nuclear export, or cytoplasmic processing of the pre-piRNAs into mature
piRNAs. Thus, gathering of pre-piRNAs into DotCom foci does not affect
their nuclear export or cytoplasmic processing. However, EJC deficiencies
result in defects in TE silencing. Although the data regarding the affects of
the knockdowns on pre-piRNA intranuclear movement and nuclear export
are convincing, this reviewer is not convinced that the authors can
conclude that defects in flam pre-piRNA intranuclear movement causes
defects in TE silencing since, as the authors admit, disruption of EJC
components could indirectly affect TE silencing.

(2) Yb-body assembly requires pre-piRNA nuclear export and occurs at
the site of pre-piRNA nuclear exit. The data document that Yb-bodies fail
to efficiently form in the absence of flam transcription (Fig. 6) as well as
when nuclear exit of pre-piRNA is aberrant (as happens upon Nxf1/Nxt1
knockdowns). The data that Yb-bodies locate at the cytoplasmic face of
nuclei, in foci close to the nuclear site of flam transcription/exit are
convincing and interesting; however, the mechanism by which Yb-body
proteins locate to pre-piRNA foci exiting the nucleus remains unknown.
Specific criticisms:

(1) Figure labeling - In the versions available to this reviewer, many of
the figures are displaced and the labels cannot be read; Fig. 3C is
particularly problematic as the entire figure has been displaced
downwards. I assume the difficulties arose during PDF conversion. Figs.
2A, 2C, S1B, and S2 each have misspellings (Y-axis). The color scheme
for Fig. S1B may not be described.

(2) Level of knockdowns - Nxt1 and Nxf1 knockdowns have quantitatively
different consequences; is this due to different knockdown efficiencies?
The levels of knockdown for all constructs should be provided.

(3) Number of nuclei viewed - The data presented in the imaging studies
are "pretty"; however, for most panels only ~2 nuclei can be viewed.
Although for most studies, quantitative results are provided, it would be
preferential to be able to view additional images as supplementary figures

Reviewer 1:

1- We apologize for having posted misaligned figures. We verified that figures are aligned in
this version.

2- We discuss the general effects of Exportin and EJC component knock-downs on the
expression of piRNA pathway factors. New experiments are presented in Supplementary Fig
5A. These verifications fit with data reported by in Malone & al. who have found that the
level of *vret*, *armi*, *yb*, *mael*, *UAP56*, *zuc* mRNA is unchanged in *rnpSI-KD* whereas *piwi*
mRNA shows a decreased level.

They also fit with results provided by Hayashi & al who reported that "splicing of most
introns is EJC-independent and no piRNA pathway mRNAs other than *piwi* seem mispliced
in Tsu- or Acn-SKD."

These two earlier studies are mentioned and a new paragraph has been added page 8/9:

"*we analyzed by RT-qPCR the impact of Mago- and RnpSI-SKD on the expression of major*
*genes involved in the somatic piRNA pathway like piwi, armitage (armi), maelstrom (mael)*
*or yb (Supplementary Figure 5). We found that the expression of none of these genes is*
*affected in these mutants except piwi whose expression decreases in Rnps1 depleted line as*
*earlier reported ((36, 37) see discussion).". It must be noted that, in the study reported here,*
*mRNA depletion may not be detected when production is mostly provided by germ cells.*

3- We now discuss the difference observed between Nxf1 and Nxt1-SKD lines. It is mainly
stated page 13 in the following paragraph :

"*..It is possible that the redundancy which has been reported between export components*
*binding nuclear pore complex may compensate depletion of Nxf1 (41, 42). Alternatively, in*
*our SKD mutant, a residual Nxf1 function could be sufficient to allow the export of enough*
*flam precursors that are processed to an almost WT amount of piRNAs (37 and 41 rpkm*
*respectively) (Figure 2E). Nxt1 plays an essential role in the export of piRNA precursors in*
*follicle cells. Several studies have reported the ability of Nxt1 to mediate nuclear mRNA*
*export by controlling the interaction of Nxf1 with components of the nuclear pore (29, 43)..."*

We have added a statistical analysis to all our results.

4- Specific functions of the EJC components are now reported in the introduction (page 4/5)
and in the discussion. The different effects of their depletion on *flam* precursor localization is
discussed. A new paragraph has been added (page 13/14)

5- Many repetitions have been deleted in the discussion.

In the new version, we now discuss conservation of EJC/exportins in other organisms and
their potential role: "...it can be anticipated that components of EJC and exportin pathways
which are well conserved throughout eukaryotes fully participate in the defense system
protecting genomes from TE invasions..." (page 15).

Concerning the role of long piRNA precursors in initiating the assembly of their processing
machineries, we added a discussion highlighting the similarities and differences observed
between piRNA precursors in the somatic lineage and the germline "*Unlike somatic cells,*
*germline has evolved a mechanism that inhibits splicing of the nascent piRNA transcript (39).*
*piRNA clusters expressed in the germline are preferentially dual-strand clusters based on*
*transcription orientation occurring from both strands. The RDC complex composed of Rhino,*
*Deadlock, Cutoff is required for the non-canonical transcription of these dual-strand piRNA*
*clusters and involves 5' end protection of nascent RNAs and suppression of transcription*
*termination. It was speculated that stalled splicing due to RDC and UAP56 differentiates*
*these piRNA precursors from mRNAs (39, 46, 47). Then, our study underlines that the non-*
*canonical transcription of dual-strand piRNA clusters in the germline and the canonical*
*transcription of uni-strand clusters in the follicle cells engage different processes both*
*leading to piRNA biogenesis."* (page 15).

6- Statistical tests have been performed, error bars and p-values added.

Minor points:

7- The sentence reporting the distinctive roles of EJC and Exportins in the abstract has been
reformulated.

8- *D. melanogaster*, Aub and Armi are now spelled out when first used.

9 – To explain why *flam* foci are not visible in some cells, it is stated in Figure 2B legend:

"... *One Z-stack of confocal image is shown. Therefore, as flam foci are not all in the same*
*focus they cannot be all visualized in every nucleus of the field.*"

10 – All axes in figures 2E and 4B are identical. Therefore, a sentence has been added in the
legend of Figures 2E and 4B: "*The Y and X-axes in .. are identical for the three graphs. X-*
*axis represents the flam 180 kb starting from its transcription start site on the left. Y-axis*
*represents the quantity of piRNAs normalized to one million genome mappable reads*"

11- Page 9: The paragraph has been reformulated: "*Since the nuclear traffic of flam*
*transcripts is disrupted in these mutants, it is expected that Yb-bodies are then juxtaposed to*
*flam transcription site. By combining DNA FISH and immunofluorescence, we confirmed that*
*a higher proportion of Yb-bodies is juxtaposed to the flam genomic locus in RnpS1 SKD line*
*compared to WT (Figures 6C and D)*".

12 - We agree that in Fig 4D and Fig 6D, the WT bars should be similar to each other. We
performed new experiments to increase the number of counts but a difference remains. Since
WT bars are always similar between several DNA/RNA FISH experiments (See Figure 2F
and 4D), it may be argued that counting can hardly be compared between DNA/RNA (Fig.
2F and 4D) and Immuno DNA FISH experiments (Fig 6D). This is potentially due to
differences in the detection of signals varying with the type of probe.

Reviewer 2

The two major flaws reported by reviewer 2 do not exist anymore.

- We added all the statistics in each Figure and supplementary Figures when needed. As
mentioned by the reviewer, all of them enforce the strength of our data.

- We are aware that genetic and molecular approaches as used here cannot discriminate
between direct and indirect roles of the genes identified. This was not clearly stated in the
former version. So, we added a paragraph and discuss this point in detail page 13: "*At*
*present, we do not know whether interaction between the flam precursor and these factors is*
*direct or indirect. EJC components may each bind flam precursor directly and independently.*
*Alternatively, some may bind directly whereas others may only associate indirectly through*
*interactions within the complex. It is also possible that these interactions occur with other yet*
*unknown factors. Moreover, we cannot exclude that through a more general effect on mRNA*
*processing and transport, EJC mutants may indirectly affect flam export and the piRNA*
*pathway.*"

1- In 2014, Murota et al reported cytoplasmic "*flam* bodies" in their study performed in
cultured cells (OSS cells). We refer to this study in the introduction. On the other hand, in
2013, we had also published data showing that, in somatic cells of the ovaries, *flam*
transcripts accumulate mostly in a nuclear focus, Dot COM, localized at the nuclear
periphery. Here we now report thousands of images (several have been added in the
supplementary figures as requested by reviewer 3), all of them coming from experiments
performed on ovaries, to analyze more closely the sub-cellular localization of *flam*
transcripts. We bring enough data to do statistics which fit with each other. We do not
believe that the observed localization patterns originate from an artifact of confocal imaging.

If they were, we should not see an increase in the proportion of cells having a nuclear
accumulation of *flam* transcripts in Exportin-SKD as we see in Figure 2A and B.
Nevertheless, to answer this comment, we added a film showing an intranuclear *flam* dot with
no ambiguity.

2 – We agree with this comment. We corrected the manuscript to present and discuss data in
light with the three categories of localization (nuclear, trans-membrane and cytoplasmic).
This has been modified for WT and depleted lines (Figure 2A and 3E). Statistics are added
for all of them so that comparisons can be made.

3- In the revised version, the statistical analysis performed for each experiment shows clearly
that we did not present, or ignored or over-interpreted differences which should not have
been. Error bar are now shown on each histogram and p values are given.

Minor points:

1- We agree that no evidence exists showing that piRNA processing occurs in Yb bodies. We
corrected the abstract to take this comment into account.

2- We focused the introduction on what is known in *Drosophila* and indicated it clearly.

3- We indeed did not refer to the recent papers reporting that primary piRNA biogenesis is
different in the somatic cells and in the germline. We believe that introducing the mechanism
of piRNA processing taking place in the germline, including phasing, is not appropriate in
this article.

To respond to this comment, we first modified the introduction to only focus on the somatic
piRNA clusters. Second, we added a full paragraph on somatic and germline differences in
the discussion page 15, in which the traffic of *flam* transcripts in the follicle cells is compared
to the one of piRNA precursors in the germline. To this purpose, we report the differences in
expression of piRNA clusters in both lineages: uni-strand vs. dual strand piRNA clusters,
canonical vs. non-canonical transcription, spliced vs. unspliced piRNA precursors. Moreover,
we also refer to Zhang et al (2014) who proposed an extrinsic control of nuage assembly and
function exerted by nuclear components, and discuss it in link with our data (page 16).

4- Reference to UAP56 has been completely modified. It is now presented and its function
more clearly reported in the introduction page 4, in the results page 8 and in the discussion
page 14 and 16.

5- We agree with this comment. We corrected the first paragraph of the results accordingly.
“Accurate quantitative analysis” has been removed.

Reviewer 3

1- We completely agree with this comment of Reviewer 3. We cannot be sure that defects in
*flam* pre-piRNA intra nuclear movement causes defects in TE silencing. We only brought this
point as an hypothesis. So, to be more clear, we left the two hypotheses in the discussion but
we now discuss that the effect can be indirect (Page 14/15) “*Why are Mago, Acn and Tsu*
*mutants unable to silence somatic TEs if piRNAs are normally produced? Several hypotheses*
*can be proposed. First, unknown proteins whose mRNA splicing is also affected by the*
*mutations could be required to produce functional piRNAs or to silenced TE at the*
*transcriptional level. Second, the traffic of flam precursors away from their transcription site*
*before leaving the nucleus may be required for the piRNAs to become functional..*”. In
addition, we removed the misleading concluding sentence.

2- To answer this comment, we added a paragraph page 16. In the germline, an extrinsic
control of nuage assembly and function exerted by nuclear components has also been
reported (38). The mechanism remains unknown but the authors proposed that Rhino and

UAP56 are connected to the nuage via the nuclear pore complex that could trigger release of
cluster transcripts in the nuage. A nuclear envelope-spanning machinery would then be
required for piRNA biogenesis. To answer to reviewer 3, we now report this study and
compare germline data to data reported here from the somatic follicle cells. We highlight
similarities and differences. This leads us to suggest that in the follicle cells, Nxt1 could play
a key role by stimulating interactions of exportins with components of the nuclear pore.

Specific criticisms:

1- We apologize for having posted misaligned figures. We verified that figures are aligned in
this version. We also corrected misspellings in figures 2A, 2C, S1B, S2 and described the
color scheme in figures when required.

2- Quantitative differences between Nxf1- and Nxt1-SKD lines are now discussed page
13: "... . *It is possible that the redundancy which has been reported between export*
*components binding nuclear pore complex may compensate depletion of Nxf1 (41, 42).*
*Alternatively, in our SKD mutant, a residual Nxf1 function could be sufficient to allow the*
*export of enough flam precursors that are processed to an almost WT amount of piRNAs (37*
*and 41 rpkm respectively) (Figure 2E). Nxt1 plays an essential role in the export of piRNA*
*precursors in follicle cells. Several studies have reported the ability of Nxt1 to mediate*
*nuclear mRNA export by controlling the interaction of Nxf1 with components of the nuclear*
*pore (29, 43)...".*

Additionally, qRT-PCR has been performed on whole ovaries of all SKD lines used in this
study.. However, since ovaries contain both germline and somatic cells, mRNA depletions
occurring in the somatic cells only can hardly or not be detected. Some differences observed
between SKD lines can be interpreted as differences in the level of mRNAs produced within
the germline. qRT-PCR experiments are provided in Supplementary figures 1C and 4A. The
affected function due to depletions is more easily detected through TE de-silencing as seen in
Supplementary Figures 1B and 4B.

3- We agree that a low number of cells were shown to illustrate each experiments, although
quantitative results on a high number of cells were provided. We thus added images as
Supplementary Figures 2, 3, 6, 7, 8, 9A, 9B, 10, 11 corresponding respectively to additional
images of Figures 2B, 2G, 3D, 4C, 4E, 5A, 5C, 6A, 6E.

We believe that this revised version of the manuscript answers each comment made by the
reviewers. We hope that it will make our manuscript suitable for publication.

Reviewers' Comments:

Reviewer #1 (Remarks to the Author)

The authors have satisfactorily addressed my concerns and comments.

Hence, I support publication of this study.

Two additional minor comments:

1. While the effects of EJC depletion are now exhaustively discussed in
the manuscript, I was still not able to find a description on how Nxf1 and
Nxt1 SKDs affect the expression of piRNA factors. Maybe a short
comment on this could still be added.

2. While error bars have now been added throughout, they are still not
present in Fig 2D. In case this is an oversight, it should be corrected.

Reviewer #3 (Remarks to the Author)

The authors have responded appropriately to the previous criticisms and
the current version is much stronger with respect to data quantitation and
alternative explanations. Overall, the data are very pretty and convincing
and the results are quite interesting. There remains concern regarding
whether flam transcript intranuclear movement to Dot COM structures is
related to the defect in TE silencing because EJC knockdowns do not form
Dot COM structures, but nuclear exit and processing of flam precursors
apparently occur efficiently. The authors suggest one intriguing possibility
that "traffic of flam precursors away from their transcription site before
leaving the nucleus may be required for the piRNAs to become
functional". Such a hypothesis is consistent with the known role of EJC
proteins to mark exiting mRNAs for downstream stability (perhaps this
should be emphasized when discussing possible models?); however, it
remains to be learned whether the defect in EJC knockdowns in TE
silencing is directly or indirectly connected to intranuclear trafficking of
flam precursors.

Reviewer #1 (Remarks to the Author):

*The authors have satisfactorily addressed my concerns and comments. Hence, I support*
*publication of this study.*

*Two additional minor comments:*

*1. While the effects of EJC depletion are now exhaustively discussed in the manuscript, I was*
*still not able to find a description on how Nxf1 and Nxt1 SKDs affect the expression of piRNA*
*factors. Maybe a short comment on this could still be added.*

Indeed we reported data for the EJC depletion but not for the Exportin complex. We now
added a sentence similar to the one reported for EJC-SKD mutants:

"The level of *piwi* and *yb* transcripts was examined in the *SKD* lines. No depletion was
detected by qRT-PCR for both genes. Nevertheless, we found that depletion of *Nxt1* and
*Nxf1* in ovarian somatic cells causes loss of Piwi nuclear localization and leads to TE
derepression (Supplementary Fig. 1A & B)²²"

*2. While error bars have now been added throughout, they are still not present in Fig 2D. In*
*case this is an oversight, it should be corrected.*

Error bars cannot be added on figure 2D as small RNA sequencing has only been performed
once. Several controls have been done:1) a qRT-PCR control has been performed on
biological and technical triplicates to verify the quality of sequenced RNAs. 2) controls have
been made through all the process to ensure the quantity and quality of RNA and the
quality of sequencing. 3) we verified that similar amount of total small RNAs have been
sequenced for each sample, and we took into account any difference by normalizing data in
Read per million of reads. 4) As an internal control, we verified that bona fide reads
homologous to exons were similar between samples: Iso1A: 24,703 rpm; Nxf1-SKD: 25,002,
Nxt1-SKD: 34,125.

Reviewer #3 (Remarks to the Author):

*The authors have responded appropriately to the previous criticisms and the current version*
*is much stronger with respect to data quantitation and alternative explanations. Overall, the*
*data are very pretty and convincing and the results are quite interesting. There remains*
*concern regarding whether flam transcript intranuclear movement to Dot COM structures is*
*related to the defect in TE silencing because EJC knockdowns do not form Dot COM*
*structures, but nuclear exit and processing of flam precursors apparently occur efficiently.*
*The authors suggest one intriguing possibility that "traffic of flam precursors away from*
*their transcription site before leaving the nucleus may be required for the piRNAs to become*
*functional". Such a hypothesis is consistent with the known role of EJC proteins to mark*
*exiting mRNAs for downstream stability (perhaps this should be emphasized when discussing*
*possible models?); however, it remains to be learned whether the defect in EJC knockdowns*

*in TE silencing is directly or indirectly connected to intranuclear trafficking of flam*
*precursors.*

A sentence has been added in discussion p. 15 :

« This hypothesis is consistent with the known role of EJC proteins to mark RNAs for
downstream stability. However it is still an open question whether the defect in TE silencing
observed in *EJC SKD* is directly or indirectly connected to intranuclear trafficking of *flam*.».